# The Difficult Balance between Ensuring the Right of Nursing Home Residents to Communication and Their Safety

**DOI:** 10.3390/ijerph18052484

**Published:** 2021-03-03

**Authors:** Matteo Bolcato, Marco Trabucco Aurilio, Giulio Di Mizio, Andrea Piccioni, Alessandro Feola, Alessandro Bonsignore, Camilla Tettamanti, Rosagemma Ciliberti, Daniele Rodriguez, Anna Aprile

**Affiliations:** 1Legal Medicine, Department of Molecular Medicine, University of Padua, via G. Falloppio 50, 35121 Padua, Italy; danielec.rodriguez@gmail.com (D.R.); anna.aprile@unipd.it (A.A.); 2Department of Medicine and Health Sciences “V. Tiberio,” University of Molise, 86100 Campobasso, Italy; 3Forensic Medicine, Department of Law, “Magna Graecia” University of Catanzaro, 88100 Catanzaro, Italy; giulio.dimizio@unicz.it; 4Department of Emergency Medicine, Gemelli, IRCCS (Scientific Institute for Hospitalization and Treatment), Catholic University of Rome-Teaching Hospital Foundation A, 00168 Rome, Italy; andrea.piccioni@policlinicogemelli.it; 5Department of Experimental Medicine, University of Campania “Luigi Vanvitelli”, via Luciano Armanni 5, 80138 Naples, Italy; alessandro.feola@unicampania.it; 6Department of Health Sciences, Section of Legal and Forensic Medicine, University of Genova, 16126 Genova, Italy; alessandro.bonsignore@unige.it (A.B.); camilla.tettamanti85@gmail.com (C.T.); 7Department of Health Sciences, Section of History of Medicine and Bioethics, University of Genova, 16126 Genova, Italy; rosellaciliberti@yahoo.it

**Keywords:** COVID-19 pandemic, nursing homes, health workers, clinical risk management, patient safety, ethics

## Abstract

The COVID-19 epidemic has had a profound impact on healthcare systems worldwide. The number of infections in nursing homes for the elderly particularly is significantly high, with a high mortality rate as a result. In order to contain infection risks for both residents and employees of such facilities, the Italian government passed emergency legislation during the initial stages of the pandemic to restrict outside visitor access. On 30 November 2020, the Italian President of the Council of Ministers issued a new decree recognizing the social and emotional value of visits to patients from family and friends. In addition, it indicated prevention measures for the purposes of containing the infection risk within nursing homes for the elderly. This article comments on these new legislative provisions from the medicolegal perspective, providing indications that can be used in clinical practice.

## 1. Introduction

The COVID-19 (SARS-CoV-2) epidemic has had a profound impact on healthcare systems worldwide [1,2,3]. The sudden need to safeguard patient rights and safety [4,5] has presented an unprecedented challenge [6,7,8]. One of the most at-risk environments has proved to be nursing homes for the elderly [9,10,11,12,13], due to greater frailty and mortality than any other caregiving context [14,15,16]. There are various contributing factors [17,18,19] such as the age of the residents [20,21]; the nursing home staff being composed mostly of nursing aides and nurses who provide basic care for residents [22]; the nonhospital nature of the organization, as it lacks the ability to provide intensive care treatment; adequate diagnostic equipment; and often suitable areas for carrying out effective and prolonged patient isolation. These factors have contributed to a high resident mortality rate as a result of the pandemic and have attracted scientific and legal attention for the purposes of devising action plans to safeguard the lives and safety of both residents and employees in such facilities [23,24,25].

One of the most critical issues is patient visitation [26,27], which, despite being of enormous emotional and existential importance for patients [28,29,30,31], may pose a real danger as a potential source of infection. During the initial stages of the epidemic, the Italian government passed emergency legislation to counteract this danger by means of a Decree of the President of the Council of Ministers (DPCM) on 8 March 2020 [32]. The law stated: “family and visitor access to residential and long-stay facilities, residential nursing homes, hospices, rehabilitation facilities and residential care homes for the elderly, both self-sufficient and non-self-sufficient, is restricted to cases authorized by the facility healthcare administration only, which is required to adopt the necessary measures for preventing potential infection transmission.”

This direction, with the same wording, was repeated in various subsequent DPCMs, including the latest dated 3 November 2020 [33]. It is based on the premise of preventing the spread of the epidemic in these facilities and focuses on the role of healthcare administrations in “restricting” family and visitor access. The needs of the residents and their family members are not taken into consideration, or if they are, they are secondary to the infection risk to residents.

The Italian National Institute of Health (ISS) Report (No. 4/2020, Rev. 2, dated 24 August 2020) entitled “Interim indications for the prevention and control of SARS-CoV-2 infection in residential social and health facilities” (ISS Workgroup: Infection Prevention and Control) [34] also stated that “family and visitor access to the facility is restricted to exceptional cases only (e.g., end-of-life situations) authorized by the healthcare administration.” The cited passage reiterates the concept of “restriction,” and the expression “exceptional cases” is emphasized in uppercase, bold and underlined text.

The ISS Report quotes but does not regulate the Interim Guidance given by the WHO, 29 June 2020 [35], Paragraph 4.2 of which is dedicated to “Administrative measures to manage visitors,” emphasizing in particular the “remote communications” element.

Of particular interest is the circular issued on 30 November 2020 [36] by the Italian Ministry of Health and sent to all healthcare agencies nationwide as the focus shifts to “safeguarding the health of residents and potential visitors.” This document, which does not interfere with the general safety provisions related to limiting the spread of COVID-19 in nursing homes drafted by the European Center for Disease Prevention and Control [37,38], represents a significant development from an ethical and patient/employee safety point of view in an environment at considerable risk.

The Ministry of Health has not released an official explanation for this new approach. However, there have been protests on the part of the family members of nursing home residents, many of whom have not been able to communicate (either in person or by telephone) with their loved ones for months due to isolation procedures, some only seeing them after their death due to COVID-19.

The ISS conducted a survey on the spread of COVID-19 in nursing and residential care homes [39], which produced interesting data regarding the restriction of family visitation. A 29-part questionnaire was sent to representatives of 3292 nursing homes (out of a total of 3417), to which 1356 facilities (41.3% of those contacted) responded.

Two of the questions were of particular interest.

“13. In line with DPCM of 8 March 2020, has patient visitation by family members/caregivers been prohibited?” All but one of the facilities that answered the question (1346 of them) replied yes. In total, 88.8% of the facilities had adopted the provision between 23 February and 9 March. There were very few exceptions to the visitation ban, permitted only in the event of serious deterioration of the resident’s health condition or final stages (end-of-life).

“14. Have alternative forms of communication with family members/caregivers been implemented?” Only six facilities admitted they had not implemented alternative forms of communication to in-person visitation. In total, 68.8% (1339) of the facilities that have implemented alternative methods of communication reported that they mostly use telephone or video calls, 19.4% only video calls and 6.5% only telephone calls and even emails on occasion. The remaining 5.5% did not provide any details.

The ISS’ optimistic view of the data notwithstanding, it should be kept in mind that 58.7% of the facilities contacted did not participate in the survey; moreover, in some facilities, the right to communication was not guaranteed, and in others, communication was restricted to telephone calls only.

The right of older people and older care home residents to social contact to communicate with the outside world is a human right. There are a lot of statements on the topic.

The fundamental reference for the right to communication is contained in Article 19 of the Universal Declaration of Human Rights, which states: “Everyone has the right to freedom of opinion and expression; this right includes the freedom to hold opinions without interference and to seek, receive and impart information and ideas through any media and regardless of frontiers.” Implied therein is the fact that everyone has the right to communicate.

Article 25 of the European Charter of Fundamental Rights concerning “The rights of the elderly” (Eustacea project, under the Daphne III Programme) states: “The Union recognizes and respects the rights of the elderly to lead a life of dignity and independence and to participate in social and cultural life.” Participation in social life necessitates implementation by means of communication.

Article 6 of the European Charter of rights and responsibilities of older people in need of long-term care and assistance concerns the “Right to continued communication, participation in society and cultural activity”; in particular, it states: “6-1. You should be made aware of and given opportunities to participate voluntarily in social life in accordance with your interests and abilities in the spirit of solidarity between generations … 6-2 You have the right to all the support necessary to enable you to communicate. You are entitled to have your communication needs and expectations taken into consideration, in whatever way these are expressed.”

On 17 December 2020, Amnesty International Italy published their research entitled “Abandoned,” a report on the violations of human rights of older residents of care homes during the COVID-19 pandemic [40]. It stated that the “authorities must guarantee care home residents … visitation policies that enable them to have regular contact with their families.”

We feel, therefore, that the Ministry’s intervention to regulate this subject matter is appropriate, if not belated.

## 2. Protecting the Right of Elderly People to Social Contact

The objective of the 30 November 2020 circular issued by the Italian Ministry of Health on the “Provisions for visitor access to residential homes, nursing homes and hospices, and indications for new admissions in the event of infected patients in the facility” is to provide more detailed operating indications than in previous legislative provisions. Said circular documents a change of cultural approach on the part of the political decisionmakers towards interactions between residents of the various types of residential and long-stay care facilities and their family members.

The document consists of a premise, five general indications and six focus points.

The premise focuses on the following situations:
(a)Residents of facilities for which the implementation of preventive measures, including “physical distancing and restrictions on social contact, has caused a reduction in interaction between individuals and in social-emotional connections, which, in a population of frail, cognitively unstable individuals, may result in further psycho-emotional degradation. This, in turn, may increase the risk of deterioration of organic diseases.”(b)Family members of residents who “have had to cope with being distant from their loved one and with the consequent difficulty in providing emotional support and assistance.”

Table 1 below contains the general indications for nationwide implementation in order to render patient visitation possible.

The objective of these five general indications is to ensure visits from relatives and volunteers “to prevent the consequences of severe isolation on the health of residents.” It is thus recognized, by means of this legal provision, that isolation, especially as regards the advanced elderly, may lead to serious consequences on cognitive function and psychophysical wellbeing [10,30,41,42,43,44,45].

Safety must be prioritized during these visits by means of appropriate protective equipment and environmental conditions. Additionally, the conditions state that all “residential facilities must make appropriate arrangements to enable each resident to connect regularly with their friends and family digitally,” especially if visits in person are not possible, and that “best practices should be developed and shared as regards managing residents’ social networks and interactions, both in person and distanced, including methods for evaluating the impact thereof in terms of efficacy and safety.” As a result, healthcare administrations must “devise a detailed plan for ensuring the option for in-person and distanced visitation with residents,” urging, for example, “hug rooms” to be set up. In any case, appropriate protocols must be developed for each hypothetical solution. The conditions repeat the need to keep entrance logs, tracing all visitors to the facility [17].

The six focus points are as follows:Perform rapid antigen tests on visitors authorized by the facility manager.Perform molecular tests to screen new resident admissions and facility staff.Suspend visitor access to residential and nursing homes when a COVID-19 case or cluster has been identified, with the opportunity to “visit patients in isolation or quarantine, in cases authorized by the facility managers” and “increasing the options for distanced interaction using various methods and ensuring family members are provided constant information on the residents’ health condition.”Suspend new admissions in residential and nursing homes when cases have been identified among the residents.Organize visitation in the event of end-of-life situations of patients affected by COVID-19.Organizational and infection prevention measures.

## 3. Managing Distanced Interaction

The new aspect of the Ministry of Health’s circular is that the premise focuses on the social and emotional aspects inherent in the residents’ right to health. This changes the approach that previous provisions in effect in Italy had taken, provisions that were focused on the role of healthcare administrations in “restricting” family and visitor access. The focus in the current document has shifted to the needs of residents and the wishes of potential visitors [46,47,48]. The premise concentrates on the potential harm and risks that restricted social interaction poses to older people and on the suffering experienced by family members due to having to limit the emotional support they can provide to elderly residents [49].

Therefore, the restrictions on visitation that healthcare administrations were previously required to implement, almost categorically, will now be implemented in line with codified rules set forth by the individual facilities [50]. In fact, the circular obligates healthcare administrations to devise a detailed plan regarding both in-person and distanced visits [51]. The ambiguity of the previous expression, “cases authorized by the facility healthcare administration,” is thus removed, as it opens the way for improvised, not rationally planned, decisions. In addition, the measures set out in the DPCM “to prevent potential infection transmission” are tempered by the principle of protection of the residents’ health, which was hitherto compromised by the lack of social interaction [29,52,53].

As regards in-person visits, protocols must be put in place and inserted as part of the above-mentioned “detailed plan.” Of interest is the topic of “distanced interaction,” which must also be provided for in the healthcare administration’s detailed plan. Distanced visits may be requested by family members, residents, or facility personnel. Therefore, distanced visitation becomes the automatic solution in all cases where the healthcare administration denies in-person visitation, as provided for in the circular. Distanced visitation may take place through appropriate structures such as windows with an intercom or through telecommunication solutions.

The second general indication has to be supplemented as it envisages the need for digital connection methods only on the part of residents. It is obvious that this section of the document is to be interpreted in that similar measures will also need to be put at the potential visitors’ disposal. The circular continues by specifying that digital interaction should be regular, with appropriate methods, frequency and duration. However, this aspect is indeterminate and clearly needs to be tailored to the needs and expectations of the residents in line with the wishes of potential visitors.

Therefore, facilities must guarantee the availability of appropriate communication methods, including digital. In other words, all potential visitors, especially those who are not in possession of the appropriate technological means for connection via the internet, must be able to avail themselves of the necessary tools and areas provided by the facility [54]. Even family members of elderly nursing home residents are often not autonomous in their use of technological communication methods and may similarly experience isolation and a lack of support from their social network. In the absence of the therapeutic benefits that the proxemic aspect of face-to-face interaction provides, telephone contact with a family member is essential for providing medical information regarding deteriorating or dramatic situations.

In view of the variety of reasons for resorting to distanced visitation, it is important that the visitation areas be set up with the third provision in mind in relation to suspending visits in person in the event of occurrent COVID-19 cases or clusters. This provision indicates that facilities must organize areas that are completely separate and manned by different staff and that facilities must “increase the options for distanced interaction using various methods” [6].

The dignity, affectivity and interactions of patients affected by COVID-19 are guaranteed even in end-of-life situations; this assumes even greater importance in hospices, considering the nature of such facilities [55,56,57]. Therefore, training must be provided for care workers to acquire the specific expertise needed to facilitate the various forms of remote communication between admitted patients and their family members, mindful that clinical updates may concern deteriorating or dramatic situations in the absence of the therapeutic benefits that the proxemic aspect of face-to-face interactions provides [58]. Figure 1 (“Steps for prevention and intervention in managing visitation in nursing homes”) shows that clinical risk management is the focal point for action plans devised by healthcare facility administrations to mitigate infectious risks within nursing homes. This enables them to care for patient needs, their right to interact with others but also the need for protection.

The Ministry of Health’s circular gives formal recognition to the principle that restricting access to nursing homes must be balanced with the right to communication. It provides general indications that can be developed, within individual residential facilities, into detailed measures in line with the specific organizations and based on recent experience acquired on an international level. This facilitates the drafting of complex measures that enable both infection prevention and the right to communication to be applied in nursing homes [59,60,61].

## 4. Clinical Risk Management

Clinical risk management represents the entirety of organized systems designed to improve the quality of healthcare services and ensure the safety of patients [62,63,64], visitors, employees and the entire organization by identifying and evaluating risks and adopting containment measures [65,66,67,68].

In relation to visiting residential care homes, the social and emotional protection of patients needs to be handled in such a way that it does not compromise the safety of patients, visitors, employees and the entire residential care home organization. In line with this principle, the document attached to the ministerial circular recommends persistence in implementing and monitoring infection prevention measures, and in handling the visitations referred to in ISS Report No. 4/2020. According to the document, these measures can be revised according to the outline in Table 2 below.

In view of the foregoing, the focus must be on the healthcare needs of residential care home workers, residents and visitors, as well as on proper clinical risk management. In the context in question, “new” clinical risk aspects have arisen, related to: (i) the balance between protecting health and safety and (ii) the attention placed on visitors as potential latent risk factors.

The first new aspect is derived from the awareness of the potential increase in risk to residents’ health as well as (to a lesser degree) to potential visitors when solely the risk of in-person visitation is emphasized and recognized, without due consideration to the psychological, social and emotional aspects of well-being, which could be compromised due to the reduction in visitation. In order to prevent said risk, no effort should be spared in facilitating in-person visits as well as the optimal conditions for distanced visitation. The revocation of in-person visits and isolation of care home residents does constitute an infection prevention measure but to the detriment of the residents’ overall well-being. Safety interventions must be adapted to accommodate patient dignity and proportionate to the objective of safeguarding health.

The second new aspect concerns the “unusual” level of attention that has to be placed on visitors due to the fact that they enter from an external environment beyond the residential home’s control. Visitors constitute an unknown variable as potential, even latent, infection risk factors. The preliminary checks (anamnesis, healthcare checks, precautionary and preventive measures) in place for in-person visits, although recommended, offer no guarantee that visitors do not represent a source of infection. Effective risk management must therefore be centered on visitor education, providing them with the appropriate means for ensuring consistently safe behavior and adopting personal protection methods during the ordinary activities of life, protecting the visitors themselves, naturally, but most of all the care home residents that they may go on to visit in the near future. Visitor education cannot be limited to information only but must be integrated with practical methodologies [69,70,71,72] in order to equip visitors with the means to engage in social interactions without the potential for infection transmission and to instill in them a constant awareness that translates into consistent behavior.

## 5. Conclusions

The circular we have examined highlights a significant change in approach to patient visitation on the part of political decisionmakers in Italy, showing that their objective is to strike the proper balance between the protection of care home residents’ right to communication and their safety. The document focuses both on residents’ healthcare needs and expectations, in relation to affectivity and support from family members, and on effective clinical risk management.

For the first time in a political–administrative document, the right of the elderly, as nursing home residents, to communication and social interaction has been expressly stated [73]. Moments of loneliness should become increasingly more seldom, in particular thanks to the requirement to enable regular opportunities for telecommunication [74]. However, family members do not always have access to the necessary technological tools, nor are they always able to travel to the facility frequently for in-person visits or to access the technological facilities placed at their disposal. Therefore, it should be acknowledged that the current situation of social disparity does not facilitate equal opportunities for all. As a result, the elderly will continue to experience moments of loneliness, despite efforts to reduce them.

The definition of health contained in the World Health Organization Constitution, stipulated in New York on 22 July 1946, states: “Health is a state of complete physical, mental and social well-being, not merely the absence of disease or infirmity.” As things stand, this definition merits consideration in relation to the social element of well-being, of which interaction and communication are a fundamental part. If these elements now depend on the availability of technological resources, access to such resources becomes an essential requirement of the social element of well-being and, therefore, of health.

Equal access to technological resources should therefore be considered in the same vein as equal access to healthcare. The same principle applies to other aspects, for example, in relation to the progressive implementation of telemedicine [75,76].

## Figures and Tables

**Figure 1 ijerph-18-02484-f001:**
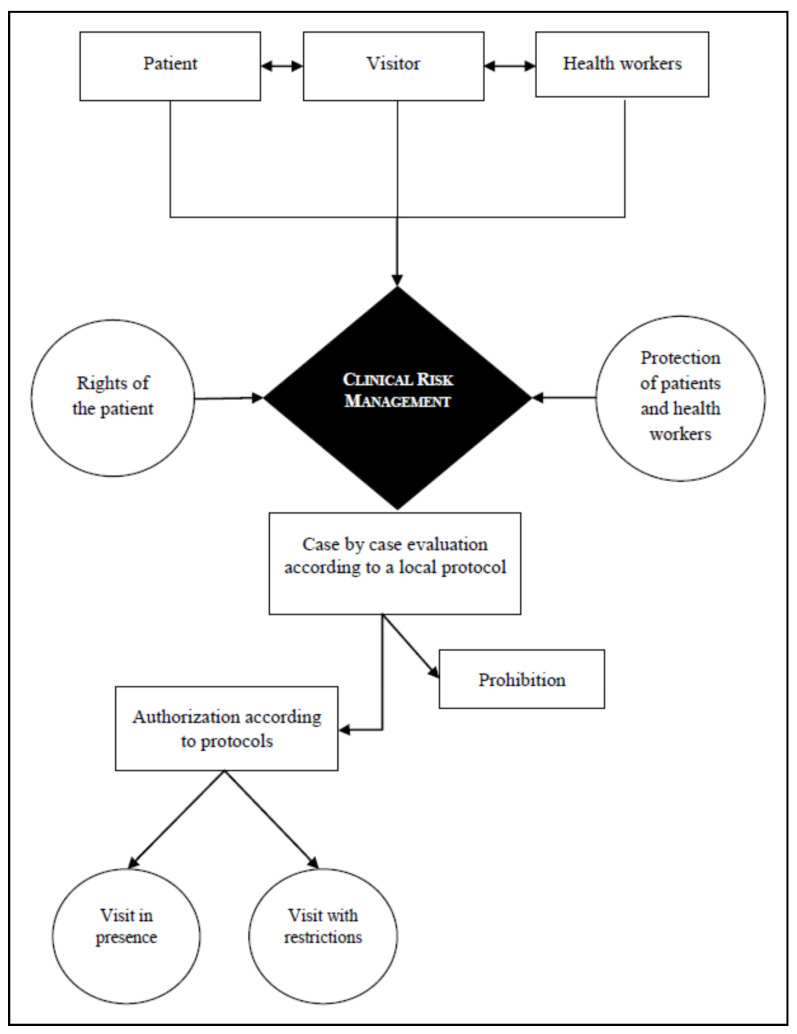
Steps for prevention and intervention in managing visitation in nursing homes.

**Table 1 ijerph-18-02484-t001:** General indications for nationwide implementation in order to render patient visitation possible.

Since social isolation and loneliness represent causes of suffering and significant risk factors to the elderly population, visits from family members and volunteers must be guaranteed in order to promote survival and physical and mental health, especially in cases of depression, anxiety and cognitive impairment/dementia, to prevent the consequences of severe isolation on the health of residents. Safety must be maintained by means of appropriate protective equipment and environmental conditions. All residential facilities must make appropriate arrangements to enable each resident to connect regularly with their friends and family digitally in order to prevent forced isolation and facilitate occasions for social and affective interaction. These tools are especially important when the epidemiological conditions of the area in which the facility is situated do not permit frequent visits in person. While complying with the appropriate risk containment measures, resumption of previously suspended care and nursing activities such as physio, speech and occupational therapy should be prioritized. In addition, having completed the necessary risk- and containment-related information/training procedures, social workers, personal assistants and volunteers should be able to resume their activities, considering the assistance they provide to residents in terms of maintaining physical and sociorelational abilities. Best practices should be developed and shared as regards managing residents’ social networks and interactions, both in-person and distanced, including methods for evaluating the impact thereof in terms of efficacy and safety. Healthcare administrations are therefore required to devise a detailed plan for ensuring the option for in-person and distanced visitation with residents. “Hug rooms” are advised so that residents in general, and especially the cognitively impaired, may benefit from safe physical contact. However, protocols must be drafted for the various hypothetical solutions, specifically as regards the required hygiene standards and personal protective equipment, for the purposes of risk containment and the safety of patients, employees, volunteers and visitors. Entrance logs must be kept, and visitation/contact in the facilities must be monitored.

**Table 2 ijerph-18-02484-t002:** Infection prevention measures and managing visitation in residential care homes.

Develop a facility-specific protocol for in-person and distanced visitation Explicit and systematic methods of entrance to the facility and ad hoc areas Healthcare training for residents and visitors Recent anamnesis (exposure to risks and disease) and healthcare checks on visitors Visitor and resident hygiene Cleaning and disinfection of in-person and distanced visitation areas Continual staff training and education Personal protective equipment Healthcare surveillance (activities of the occupational health physician, workers health and safety representatives, etc.)

## Data Availability

We choose to exclude this statement.

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
