# Peer review of "The Difficult Balance between Ensuring the Right of Nursing Home Residents to Communication and Their Safety"

_ijerph, 2021, doi:10.3390/ijerph18052484_

Round 1
Reviewer 1 Report
In my opinion, the authors set a high goal for the article. They failed to achieve the goal well. The paper is a compilation rather than an analysis. There is a lack of clear evidence for indications that can be applied to all healthcare systems worldwide.
Table I contains a lot of text, it is not clear, it does not fulfill the function of a table.
There is no explanation of Figure I in the text.
Author Response
We want to thank the reviewer for his comment which helped us improve the manuscript. We have downsized the goal of the work. With this article we want to comment on these legislative provisions which are, in our opinion, interesting and important for the risk mitigation and care of subjects in nursing homes. The structure is in fact a commentary, it did not seem possible to perform other types of study such as revision since, to date, in the literature many articles on this topic are comments or letters and do not yet have a sufficient basis of evidence, however we believe that this manuscript may be of interest to many doctors and other health professionals in addition to the health authorities who in recent months are dealing with the battle against the virus. We have also modified some passages and inserted the explanation of figure I. Unfortunately, it was not possible to summarize Table I as it contains the indications coming directly from the legislative source. We believe it is important to report them in order to make it clear what the law suggests. We hope you can understand our intent. Thank you
Reviewer 2 Report
The analyisis of legislation and technical guidance on Covid management in long term care facilities is an interesting topic. However, the paper is limited to the Italian context, without any consideration of WHO and ECDC guidance that are the core reference to the quoted ISS national guidance. Moreover there is no reference to other experience that can be found in recent literature to balance access limitation to nursing homes with care of the social and relational dimensions under lockdown. Finally, methods are not clearly described and some sentences are very difficult to uderstand due to poor English writing.
I suggest the authors to reconsider the contribution as a letter to the editor
Author Response
We want to thank the reviewer for his comments which have allowed us to improve our work. We want to indicate that the manuscript is a commentary and has the purpose of carrying out a reflection that we believe is interesting and useful for many operators in the nursing home sector. The work cannot be a systematic review as to date there are not many systematic study works but rather editorial or commentary. However, we have mentioned in the revised parts some international experiences that are present. In accordance with what you have indicated, we have taken into consideration the ECDC indications and inserted some comments. We have also examined the Italian context because from the point of view of human rights, there have been protests and a clear stance by Amnesty International and courageously there has been legislative production in this sense. We believe that given the current importance of the topic and the complexity of the topic, it is impossible to express many concepts only with a letter to the editor but that it is necessary, in accordance with the editorial indications, that the commentary form is useful. We hope you can understand our intent. Thank you
Reviewer 3 Report
1/ Aim of the paper and its main contributions
The aim of the article is - as it can be assumed - risk management in nursing homes for older people during the start of COVID-19 pandemic in Italy based on the analyses of Italian government emergency legislations. Initial legislation restricted outside visitors access, the second one gave the possibility to visit older residents by their family members and volunteers, under some conditions. First decisions didn’t take under consideration the needs of older people living in the residential care homes and their family members, next document ensured the right of older people to social contacts. The new document represented ‘a significant development from an ethical and patient/employee safety point of view in an environment at considerable risk’ (p.2). Social isolation is very dangerous for older people health and well-being. The authors pointed out that the new legislation has intensified the discussions on the new aspects, that is balance between protecting health and safety and ensuring the right to social contacts.
Areas of strength
- Important topic – the authors raise issue how to protect the health and safety of older people and, at the same time, to secure their right to contacts with their family members, in the period of COVID-19 pandemic.
- Clear argumentation based on the international literature review.
Areas of weakness
- The right of older people, older care homes residents to social contacts, to communicate with outside world (family members, friends, volunteers) is one of the human rights; it would be beneficial if the authors pay in the article more attention to the protecting human rights in case of older people. There is a lot of the European Council papers, Age Platform Europe statements on the topic.
- Add please the answer - What caused the change in the Italian administration's approach to the subject? You compared the new and previous legislation, try to add what influenced the changes.
- The article is mainly based on the Italian legislation (it should be stated in the ‘Introduction’) and recent international literature on the topic – is it possible to add more information on legislation in other countries? Is it similar to Italy?
- The article is prepared by 10 authors, so it will not be difficult to add some information on how Italian nursing homes for the elderly cope with the protecting health and safety and ensuring the right to social contacts. They will show broader context.
- Structure of the article: according to the standards indicate the method of investigation and a part ‘Discussion.’
Author Response
Reviewer 3:
1/ Aim of the paper and its main contributions
The aim of the article is - as it can be assumed - risk management in nursing homes for older people during the start of COVID-19 pandemic in Italy based on the analyses of Italian government emergency legislations. Initial legislation restricted outside visitors access, the second one gave the possibility to visit older residents by their family members and volunteers, under some conditions. First decisions didn’t take under consideration the needs of older people living in the residential care homes and their family members, next document ensured the right of older people to social contacts. The new document represented ‘a significant development from an ethical and patient/employee safety point of view in an environment at considerable risk’ (p.2). Social isolation is very dangerous for older people health and well-being. The authors pointed out that the new legislation has intensified the discussions on the new aspects, that is balance between protecting health and safety and ensuring the right to social contacts.
Areas of strength
Important topic – the authors raise issue how to protect the health and safety of older people and, at the same time, to secure their right to contacts with their family members, in the period of COVID-19 pandemic.
Clear argumentation based on the international literature review.
we want to thank the reviewer for his comments which have allowed us to improve our work. we too believe that the subject of this commentary is important and current for the moment in which we are living and we think that making this change known internationally in Italian government decrees is important and of interest also for readers of many countries.
Areas of weakness
The right of older people, older care homes residents to social contacts, to communicate with outside world (family members, friends, volunteers) is one of the human rights; it would be beneficial if the authors pay in the article more attention to the protecting human rights in case of older people. There is a lot of the European Council papers, Age Platform Europe statements on the topic.
We especially thank the reviewer for this comment, we strongly agree on the issue of human rights. In accordance with these indications, on p. 3 we have included references to the Universal Declaration of Human Rights, the Europena Charter of Foundamental Rights and an investigation carried out by Amnesty International. We believe these new indications can better enhance the fundamental rights of patients in nursing homes.
Add please the answer - What caused the change in the Italian administration's approach to the subject? You compared the new and previous legislation, try to add what influenced the changes.
We do not know exactly what led the government to produce this change in the legislative decree, however we think that the action of associations of patients' relatives and some inquiries such as that of Amnesty international has sensitized the government and brought about the correction of the previous provision. On p. 2 we tried to explain it, we cited the Amnesty International investigation, the protests of the family members, the results of a survey carried out throughout the country.
The article is mainly based on the Italian legislation (it should be stated in the ‘Introduction’) and recent international literature on the topic – is it possible to add more information on legislation in other countries? Is it similar to Italy?
We thought about doing this job, in fact we cite the few articles published on the activities carried out in other countries. However, we considered that our article is limited to being a comment on national provisions and not a review of the literature also because at the moment there are very few studies carried out, there are rather editorials, comments and letters. For these reasons it becomes difficult to insert further foreign examples, moreover some countries have not produced legal documents but have relied on scientific societies and hospitals and have not had legislative production to indicate.
The article is prepared by 10 authors, so it will not be difficult to add some information on how Italian nursing homes for the elderly cope with the protecting health and safety and ensuring the right to social contacts. They will show broader context.
To achieve the indicated objective we have mentioned important data from a national survey (p. 2-3), we believe that this work shows what has been done in Italy, furthermore on p. 6 we have tried to clarify the useful information coming from the Italian experience.
Structure of the article: according to the standards indicate the method of investigation and a part ‘Discussion.’
On the structure of the work, we have had indications from the editor that being a commentary and not a revision, the structure is free and for clarity different titles can be used with respect to discussion and materials and methods. In fact it is a narrative comment and not a systematic review which would be impossible given the scarcity of research articles on the subject. if however you think it necessary to change some titles we can do it. Thanks again for your comments which we appreciate very much, We hope you can understand our intent. Thank you
Round 2
Reviewer 1 Report
I consider the topic of the work very important. I recommend following up with research activities.
Reviewer 3 Report
I accept all improvements of the article proposed by the authors.